# A putative membrane-associated YdjB-like protein from *Salmonella enterica* exhibits a non-canonical ACT-like fold

Ju Hyeong Kim[1,2☯], Yong Jun Kang[1,2☯], Young Woo Kang[1,2], So Eun Park[1,2], Hyo Been Jin[1,2], Eunmi Hong[3], Hyun Ho Park[1,2]*

1 College of Pharmacy, Chung-Ang University, Seoul, Republic of Korea, 2 Department of Global Innovative Drugs, Graduate School of Chung-Ang University, Seoul, Republic of Korea, 3 New Drug Development Center, Daegu-Gyeongbuk Medical Innovation Foundation, Daegu, Republic of Korea

☯ These authors contributed equally to this work
* xrayleox@cau.ac.kr

## Abstract

The Aspartokinase-Chorismate mutase-TyrA (ACT) domain is a conserved regulatory module found in a wide range of proteins involved in diverse biological processes, including amino acid metabolism, signal transduction, and cytoskeletal organization. Despite extensive studies, the structural and functional diversity of ACT domains continues to expand. Here, we present the high-resolution crystal structure of an 18 kDa YdjB-like protein from Salmonella enterica (seYdjB-like protein), a conserved but previously uncharacterized ACT domain-containing protein. Structural analysis reveals a novel ACT-like fold distinct from canonical ACT domains, despite superficial resemblance to aspartokinase. Interestingly, seYdjB-like protein forms a stable dimer in solution and is predicted to be associated with the membrane, suggesting functional divergence from typical cytosolic ACT domain proteins. This structure represents an alternative form of the ACT domain, expanding our understanding of ACT domain architecture and its functional potential. Given its conservation across bacterial species such as E. coli and S. enterica, this putative membrane-associated ACT-like protein may play an important but previously unrecognized role in bacterial physiology. Our results identify this 18 kDa ACT-like domain containing YdjB-like protein as a unique dimeric membrane protein featuring a non-canonical ACT-like fold.

## Introduction

The Aspartokinase-Chorismate mutase-TyrA (ACT) domain is a conserved structural motif found in a diverse array of proteins, ranging from bacterial signaling proteins to eukaryotic cytoskeletal proteins [1]. This motif consists of around 70~80 amino acids and forms a compact fold comprising of four β-strands and two α-helices arranged in a βαββαβ order [2–4]. The first structural information of ACT domain was revealed

**Data availability statement:** All relevant data are within the paper. The atomic coordinates and structure factors have been deposited in the Protein Data Bank (PDB) under accession number 9VP8 (https://www.rcsb.org/structure/9VP8).

**Funding:** This study was supported by the National Research Foundation of Korea (NRF) grant funded by the Korea government (MSIT) (RS-2025-02316334 and RS-2024-00344154). The funders had no role in study design, data collection and analysis, decision to publish, or preparation of the manuscript. There was no additional external funding received for this study.

**Competing interests:** The authors have declared that no competing interests exist.

in 1995 with the crystal structure of D-3-phophoglycerate dehydrogenase [5]. Since then, sequence analysis has revealed that a diverse group of proteins involved in amino acid and purine metabolism, and regulated by specific amino acids, contain this domain [6,7]. The name ACT originates from the first letters of three enzymes—Aspartokinase, Chorismate mutase, and TyrA—that contain this distinctive domain.

The ACT domain has been extensively studied over the past few decades, and much has been learned about its structure and function. One of the key features of the ACT domain is its ability to mediate protein-protein interactions through a variety of mechanisms, including self-dimerization, binding to small molecules such as $Ca^{2+}$ and lipid, and binding to other proteins [8]. This versatility allows proteins containing the ACT domain to play a wide range of biological processes including cell signaling, transcriptional regulation, and even cytoskeletal organization, reflecting their functional versatility [6,8–10]. In several cases, tandem or multiple ACT domains are present within a single protein, enabling complex regulatory behaviors through cooperative or independent ligand binding [1,4].

Increasing diversity of ACT domain and ACT domain containing proteins in tertiary and quaternary structures and mode of ligand binding is one of the major interests in the field of ACT domain research. Despite the considerable progress made in understanding the ACT domain and ACT domain containing proteins, more diverse functions with various mechanisms are still uncovering.

ACT domain containing YdjB-like protein, composed of around 160 amino acid residues (~18kDa) and not characterized, has been uploaded in various protein information servers including UniProt [11]. This particular protein sometimes is called as YdjB-like protein because nicotinamidase (previously called as YdjB) contains this ACT-domain that are similar with that of ACT-domain containing YdjB-like protein. This 18 kDa YdjB-like protein is conserved in various bacterial species including *Escherichia coli* and *Salmonella enterica*. *S. enterica* is a *Gram*-negative bacterium that can cause foodborne illness producing a range of symptoms in humans, including fever, diarrhea, abdominal cramps, and vomiting [12]. In most cases, these symptoms are self-limiting and resolve within a few days without medical treatment. However, in some cases, the infection can lead to more severe illness, particularly in young children, elderly individuals, and people with weakened immune systems.

Overall, the precise function of 18kDa YdjB-like protein in bacterial physiology remains largely unknown, but its conservation across bacterial species suggests that it plays an important role for bacteria survival. In this study, we report the high-resolution crystal structure of an 18 kDa YdjB-like protein from *S. enterica* (hereafter referred to as seYdjB-like protein), revealing a novel and distinct variant of the ACT domain that has not been previously described. Although the overall architecture of the seYdjB-like protein superficially resembles that of aspartokinase, detailed structural analysis reveals that it adopts a completely different fold. In addition, we reveal that seYdjB-like protein is a membrane protein and forms a dimer in solution. Based on this observation, we finally concluded that uncharacterized 18 kDa YdjB-like protein might be novel dimeric membrane protein that contains a unique ACT-like fold.

## Materials and methods

### Protein expression and purification

The majority of experimental procedures, including protein purification, crystallization, structure determination, and SEC-MALS analysis, closely followed the protocols established in our previous studies [13].

The N-terminal region deleted seYdjB-like protein (amino acid from 35 to 162) (GenBank ID: AAF68933) in *S. enterica* was synthesized by Bionics (Seoul, Republic of Korea) and cloned into a pET28a expression vector using NdeI/XhoI restriction sites. The plasmid encoding seYdjB-like protein was transformed into E. coli BL21 (DE3) cells and plated on lysogeny broth (LB) agar plate. The plate was then incubated in the 37 °C incubator. A single colony generated on the agar plate was picked and inoculated in 10 mL LB medium containing 50 µg/mL kanamycin for 16 h in 37 °C shaking incubator, after which the cells were transferred in 1 L LB medium for large culture. When the optical density value at 600 nm reached approximately 0.7, 0.25 mM isopropyl β-D-1-thiogalactopyranoside was added to the medium for target gene induction. The induced cells were further cultured in the 20 °C shaking incubator for 18 h and harvested by centrifugation. The collected cells were resuspended in 30 mL of lysis buffer (20 mM Tris-HCl pH 8.0, 500 mM NaCl, and 25 mM imidazole) and were disrupted by sonication on ice with four bursts of 5 s each and a 25 s interval between two bursts after adding 1 mM phenylmethanesulfonyl fluoride (Sigma-Aldrich, St. Louis, MO, USA). The lysed cells were centrifuged at 10,000×g for 30 min at 4 °C to separate supernatant from cell debris. The supernatant was mixed with nickel nitrilotriacetic acid (NTA) resin (QIAGEN, Hilden, Germany) through gentle agitation for 2 h at 4 °C. The mixture of supernatant and NTA resin was transferred to a gravity flow column and washed with 50 mL of washing buffer (20 mM Tris-HCl pH 8.0, 500 mM NaCl, and 60 mM imidazole). Then, 600 µL of elution buffer (20 mM Tris-HCl pH 8.0, 500 mM NaCl, and 250 mM imidazole) was loaded onto the column to elute the target protein. The resulting eluate was subjected to size-exclusion chromatography (SEC) using an ÄKTA Explorer system (GE Healthcare, Chicago, IL, USA) equipped with a 24 mL Superdex 200 Increase 10/300 GL column (GE Healthcare) pre-equilibrated with SEC buffer (20 mM Tris-HCl pH 8.0 and 150 mM NaCl). The main peak fractions were collected and concentrated to 7.6 mg/mL, flash-frozen in liquid N2, and stored at −80 °C until further crystallization.

### Crystallization and data collection

seYdjB-like protein was crystallized by the hanging-drop vapor diffusion method. The concentrated 1 µL protein sample to 7.6 mg/mL in SEC buffer was mixed with an equal volume of the reservoir solution. The mixed droplet was allowed to equilibrate with 300 µL of the mother liquor in an 20 °C incubator. The initial crystal was produced from a reservoir solution comprising 20% (W/V) PEG 8000, 0.1 M MES pH 6.0, and 0.2 M Ca(Oac)$_2$. The crystal appeared in 4 days and grew to a maximum size of 0.1 × 0.2 × 0.1 mm. For data collection, the crystal was frozen in a N$_2$ stream at −178 °C in 10% glycerol cryoprotectant condition. The X-ray diffraction data were collected at the 5C beamline at the Pohang Accelerator Laboratory (Pohang, Republic of Korea). The diffraction data were indexed, integrated, and scaled using the HKL-2000 program [14].

### Structure determination and refinement

The Phaser program [15] of the PHENIX package [16] was used for performing molecular replacement (MR) to determine the structure of seYdjB-like protein. The modelled structure generated by alphafold 2 was used as search model for MR. Model building and refinement were performed using Coot [17] and phenix.refine tools from the PHENIX package [16]. The quality of the model was validated using MolProbity [18]. All the structural representations were generated using PyMOL [19].

### SEC-MALS analysis

The absolute molar weight of seYdjB-like protein in solution was determined by SEC-MALS. The purified target protein was filtered using a 0.2 µm syringe filter and injected into a ÄKTA Explorer system equipped with a 24 mL Superdex

75 10/300 GL column (GE Healthcare) pre-equilibrated with SEC buffer. The chromatography system was coupled to a three-angle light scattering detector (mini-DAWN EOS) and a refractive index detector (Optilab DSP) (Wyatt Technology, Santa Barbara, USA). The mobile phase buffer was pumped at a rate of 0.6 mL/min. The absolute molecular mass was assessed using the ASTRA program (Wyatt Technology). A bovine serum albumin protein was used as the reference value.

## Results and discussion

### Sequence analysis and purification of seYdjB-like protein

Nicotinamidase made from *ydjB* gene (currently called *pncA*) is different from uncharacterized YdjB-like protein. First, the size of protein is around 160 amino acids in the case of YdjB-like protein, whereas the YdjB nicotinamidase consisted of around 220 amino acids. Since the 18 kDa YdjB-like protein in any bacteria has not been characterized, we initially analysed the sequence of this protein. The representative YdjB-like protein from *S. enterica* (seYdjB-like protein) composed of 162 amino acids. Domain prediction severs, SMART [20] and UniProt [11], indicated that seYdjB-like protein contains two domains, transmembrane domain (TM) (From M1~M35) and ACT domain (from F36~A100), indicating that seYdjB-like protein might be membrane protein that contains ACT domain (Fig 1a). To confirm the possibility of membrane protein of seYdjB-like protein, TM prediction sever, TMHMM [21], was hired and the sequence of seYdjB-like protein was further analysed. This prediction showed that N-terminal part of seYdjB-like protein has a transmembrane domain (Fig 1b). Based on this sequence analysis, we predicted that seYdjB-like protein might be a membrane protein (Fig 1c). Since transmembrane region can be obstacle for purification of target protein for structural study, we made an expression construct that contained only extracellular part of seYdjB-like protein (F36~K162) (Fig 1a and 1c)

To characterize the function and structure of 18 kDa YdjB-like protein family, we obtained the soluble seYdjB-like protein by purification using quick two step chromatography, affinity chromatography followed by size-exclusion chromatography (SEC). In the SEC, since 18 kDa seYdjB-like protein eluted at approximately 17 mL, which is the midpoint between the elution volumes of ovalbumin (44 kDa size marker) and myoglobin (17 kDa size marker), we assumed that seYdjB-like protein exists in a dimeric state in solution (Fig 1d and 1e). The target protein was successfully crystallized and the 1.92 Å high-resolution crystal structure of seYdjB-like protein was solved with the MR phasing method. The final structural model was refined to $R_{work} = 19.06\%$ and $R_{free} = 21.80\%$. The crystallographic and refinement statistics are presented in Table 1.

### Overall structure of seYdjB-like protein

In the crystallographic asymmetric unit (ASU), one molecule was found and the final structural model was built from amino acid 36–162. Upon analyzing the structure of the seYdjB-like protein, it was composed of four α-helices (α1–α4) and seven β-strands (β1–β7), with the four α-helices surrounding the central seven β-sheets (Fig 2a and b). The structure of this protein clearly consists of two distinct domains. One domain spans from amino acid 36–101 that was predicted as a ACT domain, while the other extends from amino acid 102 to the final amino acid, 162. Based on this arrangement, the domain located in the N-terminal part of the protein was named the ACT-like domain, and the domain in the C-terminal part was named the C-terminal domain (CTD) (Fig 2c).

*B*-factor analysis showed that several loops, including the β2-β3 and α3-β4 connecting loops, exhibited higher *B*-factors (average, 42.8 Å²) than other regions (average, 20.2 Å²) (Fig 2d). Given the importance of surface characteristics in uncovering the functions of proteins with unknown roles, we examined the surface polarity of the seYdjB-like protein. Analysis of the electrostatic surface indicated a complex arrangement of positive, negative, and neutral charges, featuring a distinct deep groove situated between the ACT-like domain and CTD (Fig 2e).

Following the determination of the seYdjB-like protein structure using the AF2-predicted model, we assessed the similarity between this model and the experimentally obtained structure. The superposition analysis suggested that the backbone of the predicted model was comparable to the experimental crystal structure, except for the incorrectly modeled

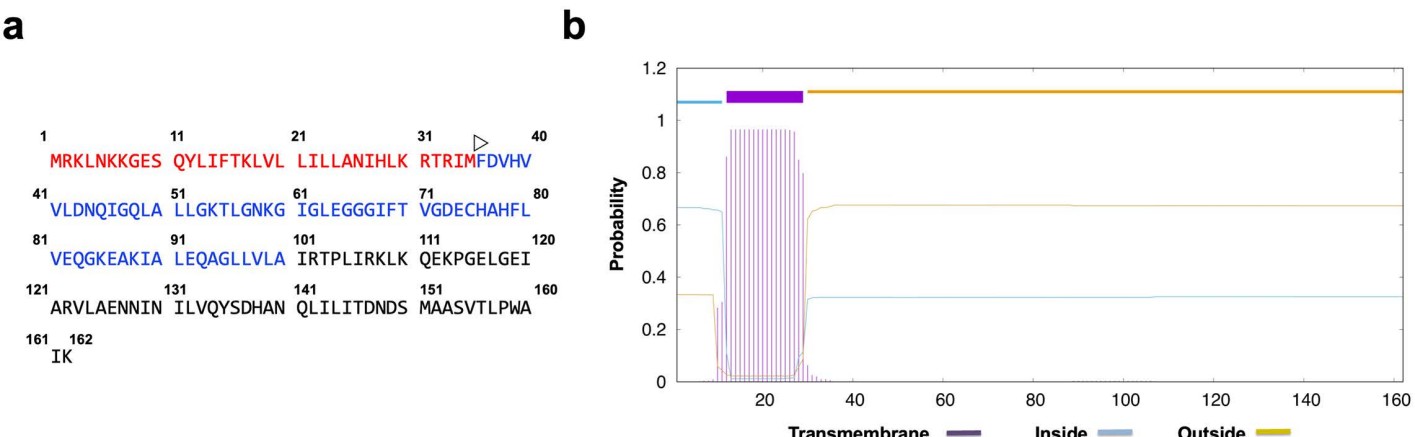

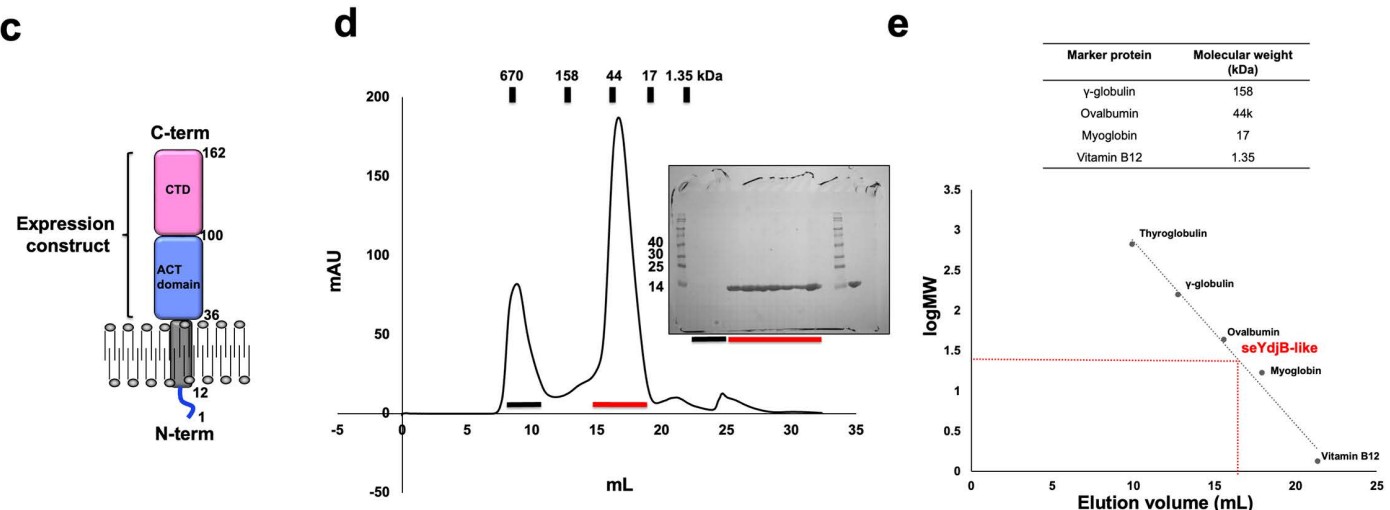

**Fig 1. Characterization of seYdjB-like protein. (a)** Amino acid sequence of seYdjB-like protein. Red and blue indicate TM domain and ACT domain, respectively. White open arrow indicates the starting point for our expression construct. **(b)** Result of TMpred website that predict the membrane protein (https://embnet.vital-it.ch/cgi-bin/TMPRED_form_parser). **(c)** Predicted domain composition of seYdjB-like protein. CTD indicates C-terminal domain. **(d)** Profile of the size-exclusion chromatography (SEC) performed to purify seYdjB-like protein. The eluted positions of standard size markers are shown above the profile. The SDS-PAGE gel loaded with the peak fractions is provided to the left of main peak. Loaded fractions corresponding to the bands on the SDS-PAGE gel are indicated by black and red arrows. M indicates the size marker. **(e)** Calibration curve of elution volumes from SEC, showing the logarithmic relationship between molecular weight and elution volume. The experimental elution volume of seYdjB-like protein is plotted for comparison.

β2 and β5-α3 connecting loop region (Fig 2f). Despite this, the overall predicted structure was in close alignment with the crystal structure.

## seYdjB-like protein forms a dimer in solution

Based on the SEC results, the seYdjB-like protein appears to predominantly exist as a dimer; however, to more accurately determine its state in solution, we conducted a multi-angle light scattering (MALS) experiment to obtain a more precise measurement of its molecular mass. This MALS analysis showed that the absolute molecular mass of seYdjB-like protein in solution was 42.8 kDa (6.7% fitting error) (Fig 3a). Given that the theoretical molecular weight of monomeric seYdjB-like

**Table 1. Data collection and refinement statistics.**

| Data collection | |
|---|---|
| Space group | $P\,4_3\,2_1\,2$ |
| Unit cell parameter | |
| $a$, $b$, $c$ (Å) | 42.88, 42.88, 140.60 |
| $\alpha$, $\beta$, $\gamma$ (°) | 90, 90, 90 |
| Resolution range (Å)[1] | 29.64–1.92 (1.989–1.92) |
| Total reflections | 270874 |
| Unique reflections | 26298 |
| Multiplicity | 25.3 (25.2) |
| Completeness (%)[1] | 99.92 (99.90) |
| Mean $I/\sigma(I)$[1] | 50.97 (28.25) |
| $R_{merge}$ (%)[1,2] | 5.235 (10.67) |
| Wilson $B$-factor (Å²) | 20.50 |
| **Refinement** | |
| Resolution range (Å) | 29.64–1.92 |
| Reflections | 26298 |
| $R_{work}$ (%) | 19.06 (18.10) |
| $R_{free}$ (%) | 21.10 (20.45) |
| No. of molecules in the asymmetric unit | 1 |
| No. of non-hydrogen atoms | 1091 |
| Macromolecules | 979 |
| Solvent | 112 |
| Average $B$-factor values (Å²) | 22.15 |
| Macromolecules | 21.21 |
| Solvent | 30.45 |
| Ramachandran plot: | |
| Favored/ allowed/ outliers (%) | 99.21/ 0.79/ 0.00 |
| Rotamer outliers (%) | 0.00 |
| Clashscore | 2.00 |
| RMSD bonds (Å)/ angles (°) | 0.004/ 0.77 |

[a]Values for the outermost resolution shell are indicated in parentheses

[b]$R_{merge} = \sum_h \sum_i |I(h)_i - <I(h)>| / \sum_h \sum_i I(h)_i$, where $I(h)$ is the observed intensity of reflection h and $<I(h)>$ is the average intensity obtained from multiple measurements.

protein with the C-terminal His-tag is 18.8 kDa, the measured mass corresponds to a dimeric form. Therefore, based on SEC and MALS data, we conclude that seYdjB-like protein forms a dimer in solution.

Based on these results, we further examined the crystallographic packing to identify a molecule that could plausibly form a dimer with Molecule A in ASU (Fig 3b). Through this analysis, we identified a symmetric partner, Mol A′, that forms an interface with Mol A (Fig 3c). Further examination of the protein–protein interaction interface between Mol A and Mol A′ using PDBePISA strongly suggested that these two molecules are likely to form a biologically relevant dimer. PDBePISA revealed a complex formation significance score (CSS) of 1 for the Mol A/Mol A′ dimer (the score ranges from 0 to 1, with higher values indicating greater biological relevance), suggesting that the dimeric interface may be physiologically relevant (Fig 3d). A total of 98 residues (49 from each monomer) are involved in the PPI formation, with a total buried surface area of 1727.0 Å², accounting for 23.2% of the total molecular surface area (Fig 3d). The dimerization is stabilized by

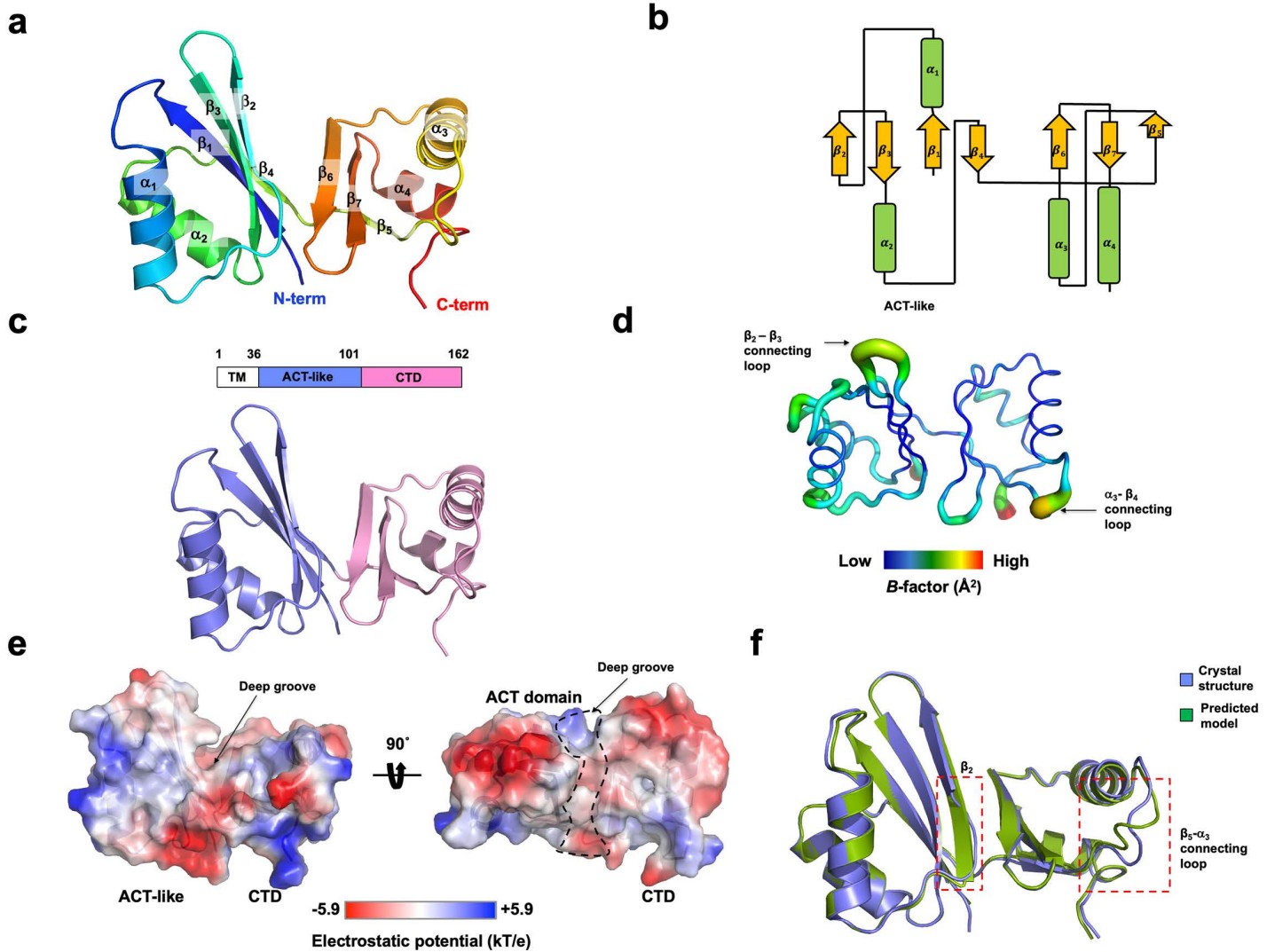

**Fig 2. Structure of seYdjB-like protein. (a)** Cartoon representation of seYdjB-like protein. A rainbow color scheme was used for tracing the path from N- to C-terminus. Helices and sheets are labeled with α and β, respectively. **(b)** Topology diagram generated based on the structure of seYdjB-like protein. **(c)** Cartoon representation showing the domain structure of seYdjB-like protein. **(d)** Putty representation showing the *B*-factor distribution. Rainbow colors range from red to violet to reflect *B*-factor values for visualization. Higher *B*-factor regions are indicated by black arrows. **(e)** Surface electrostatic potential of seYdjB-like protein, depicted using a color gradient from negative potential (red, −5.9kT/e) to positive potential (blue, +5.9kT/e). **(f)** Superimposition of experimentally resolved seYdjB-like protein structure with the AlphaFold3 predicted model. Structurally different regions were indicated by red-dot box.

hydrophobic interactions and 18 hydrogen bonds (H-bonds) across the interface. Nine residues from each molecule (N44, I46, I68, T70, N130, L132, Q134, H138, and N140) contribute to H-bond formation (Fig 3e).

## Structural comparison with structural homologs revealed that the seYdjB-like protein adopts an entirely different fold

To investigate the identity of the seYdjB-like protein, we performed a structural homologue search using DALI server (Holm & Sander, 1995). As expected, the top structural homologs identified were aspartokinases (hereafter called AK)

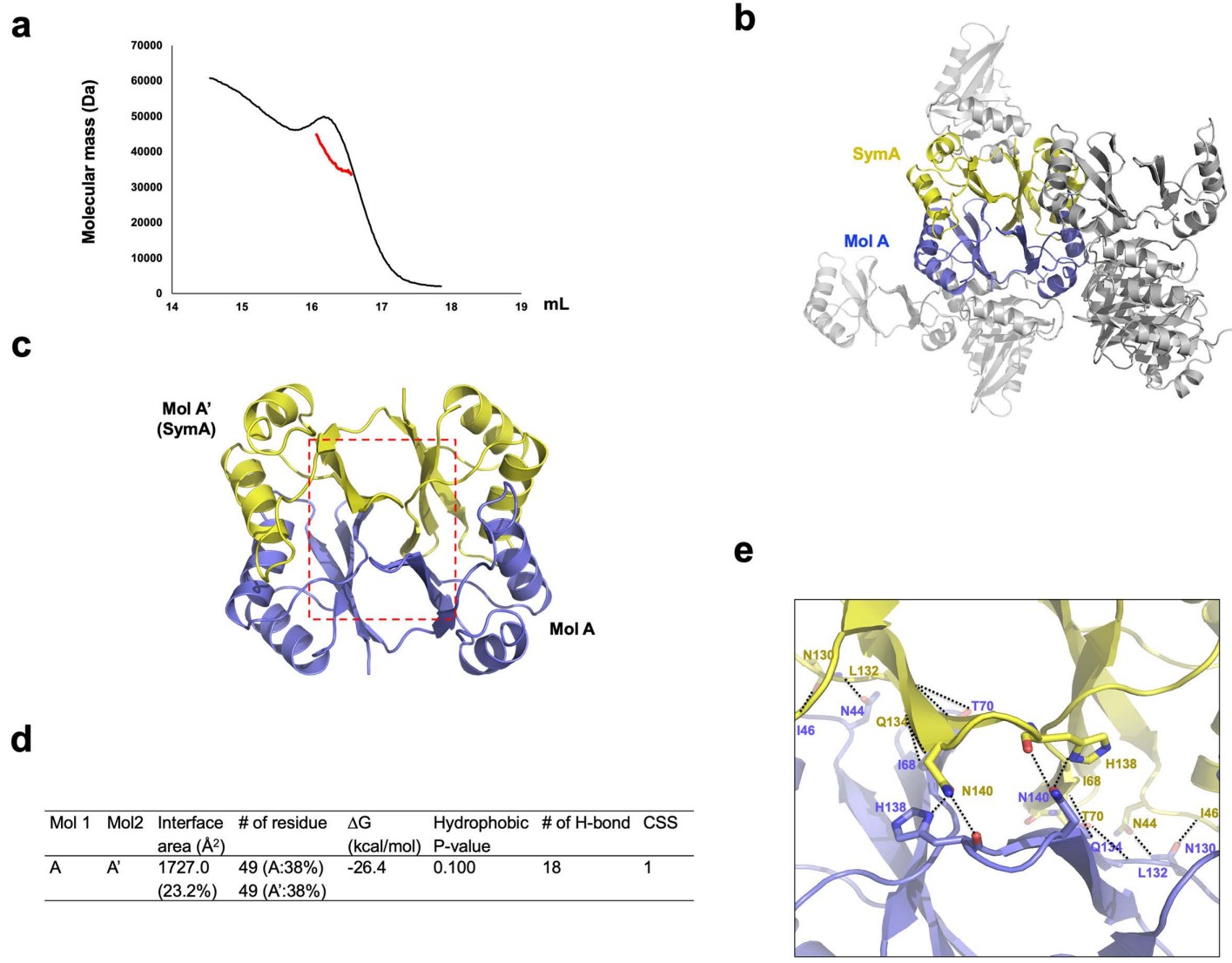

**Fig 3. Dimeric structure of seYdjB-like protein and its interface analysis. (a)** Multi-angle light scattering (MALS) profile corresponding to the main SEC peak of seYdjB-like protein. The experimental molecular weight determined by MALS is shown as a red line. **(b)** Symmetry analysis of crystallographic packing, illustrating the MolA/SymA dimeric from identified in the asymmetric unit. **(c)** A separate view of the dimer analyzed in panel **(b)**. The area shown in the close-up view in panel (e) is marked with a red-dot box. **(d)** Summary table of dimer interface properties analyzed using the PISA server. **(e)** Magnified view of the PPI regions from panel **(c)**, highlighting hydrogen bonds (black dashed lines).

from two bacterial species, AK from *Corynebacterium glutamicum* (hereafter called cgAK) [22] and AK from *Pseudomonas aeruginosa* (hereafter called paAK) [23]. Despite this similarity, the Z-scores ranged around 10–11, and the amino acid sequence identity was relatively low (11–17%), indicating that while there is some structural resemblance, the seYdjB-like protein is not highly similar at either the structural or sequence level (Table 2). However, direct structural superposition revealed that the three proteins share a high degree of structural similarity (Fig 4a). Pairwise superposition of each AK with the seYdjB-like protein showed that the RMSD values were 2.3 Å for cgAK (Fig 4b) and 2.2 Å for paAK (Fig 4c), indicating a strong structural resemblance despite the low sequence identity.

**Table 2. Structural similarity search using DALI.**

| Proteins (accession numbers) | Z-score | RMSD (Å) | Identity (%) |
|---|---|---|---|
| Aspartokinase from *Corynebacterium glutamicum* (2DTJ) | 11.0 | 2.7 (120/164) | 17 |
| Aspartokinase from *Corynebacterium glutamicum* (3AB2) | 10.7 | 2.3 (117/153) | 16 |
| Aspartokinase from *Pseudomonas Aeruginosa* (5YEI) | 10.6 | 2.2 (116/153) | 11 |

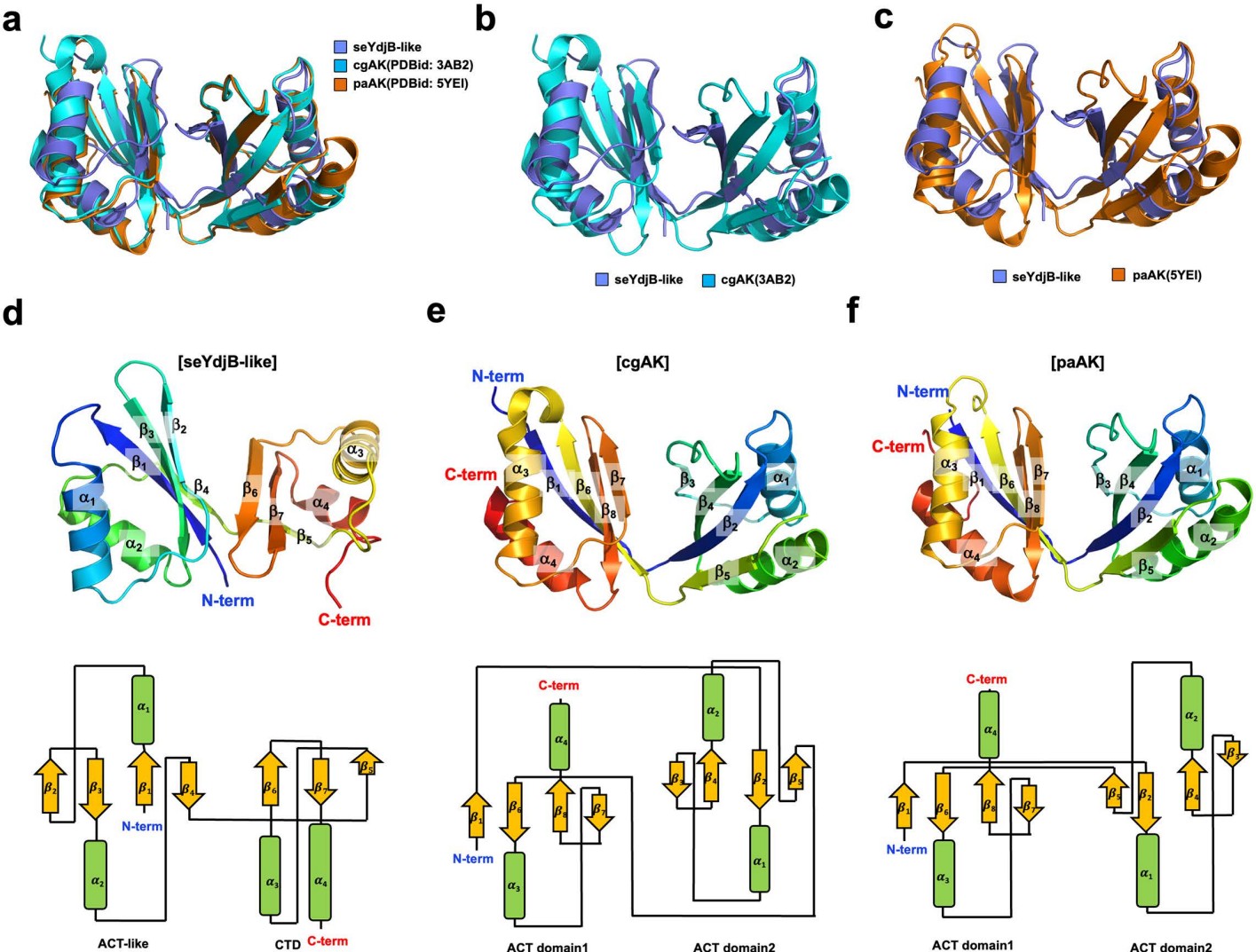

**Fig 4. Structural comparison of seYdjB-like protein with structural isoforms from different species. (a)** Structural superimposition of monomeric seYdjB-like protein with the top two structures that were identified as structural homologues by the DALI server. cg: *Corynebacterium glutamicum*, pa: *Pseudomonas aeruginosa* capsulatus. The PDB IDs are provided in the parentheses. **(b and c)** Pairwise structural superimposition of seYdjB-like protein (blue) with cgAK (cyan) (b) and paAK (orange) **(c)**. (d~f) Topology comparison of seYdjB-like protein (d) with cgAK **(e)**, and paAK **(f)**.

Interestingly, detailed structural analysis revealed that the seYdjB-like protein possesses an atypical ACT domain–like fold. Like the canonical ACT domain, the ACT-like fold of seYdjB-like protein is also composed of two α-helices and four β-sheets (Fig 1d). However, while the ACT-like fold is formed as a continuous structure extending from the N-terminus, the ACT domain of. AK is formed by the β-sheet from the N-terminus coming together with structural elements from the C-terminus (Fig 4b and c). In addition, in contrast to other AK proteins, which typically contain two ACT domains, the seYdjB-like protein is composed of one ACT domain–like fold at the N-terminus and a C-terminal domain (CTD) that also adopts an ACT domain–like shape (Fig 4d~f). Structural alignment showed that the N-terminal ACT domain-like fold of the seYdjB-like protein aligns with the ACT domain 1 of AK, which is C-terminal part of AK, indicating an inverted structural arrangement.

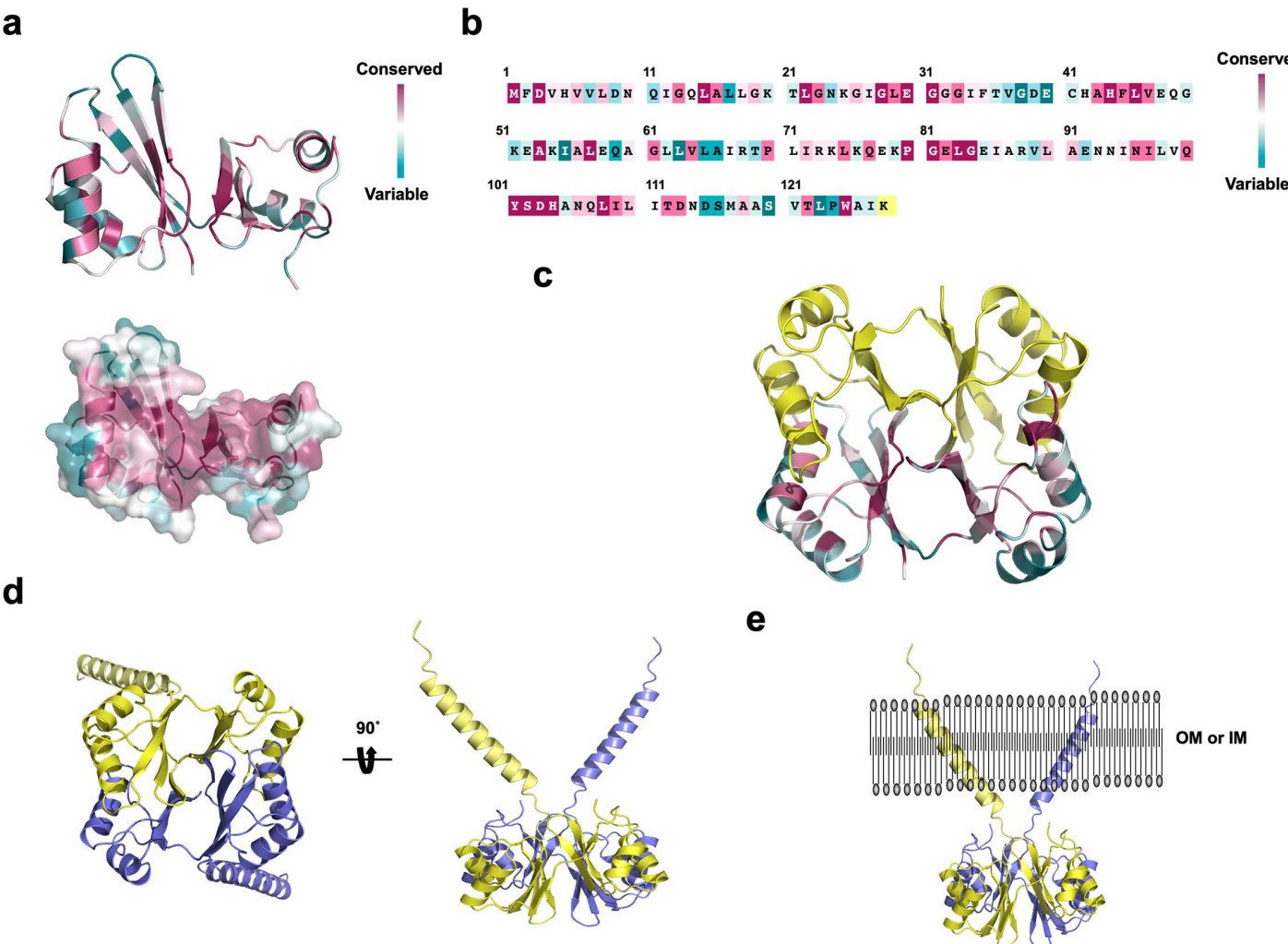

**Fig 5. Final putative dimer model of the full-length YdjB-like protein. (a)** Graphic representation of seYdjB-like protein colored relative to the amino-acid sequence conservation degree generated by the ConSurf server. **(b)** Linear representation of the amino acid sequence of seYdjB-like protein annotated with ConSurf conservation scores. **(c)** Cartoon representation of the dimeric seYdjB-like protein model colored by residue conservation **(d)** Predicted full-length dimeric structure of seYdjB-like protein including the TM domain. **(e)** Proposed membrane localization model of the dimeric YdjB-like protein.

Taken together, although the seYdjB-like protein shares overall structural features with YdjB proteins—hence its annotation as "YdjB-like"—it exhibits a unique organization in which similar folds are arranged in a different directional configuration. This suggests that the seYdjB-like protein, despite structural resemblance, represents a distinct protein with a different evolutionary trajectory or functional adaptation.

### YdjB-like protein in Salmonella enterica might be a novel dimeric membrane protein

To gain clues about the potential function and mechanism of this seYdjB-like protein, we used ConSurf to analyze evolutionarily conserved amino acids that are likely important for its function [24]. The results revealed that the central region of the seYdjB-like protein is highly conserved, suggesting that this region may serve as the enzymatic active site (Fig 5a and b). Given that the seYdjB-like protein exists as a dimer, we examined the location of this conserved region within the dimer structure and found that it is also positioned at the center of the dimer interface (Fig 5c). Based on these findings, we propose that the seYdjB-like protein adopts an L-shaped structure with an active site located at its center, and that in the dimeric form, it possesses two active sites situated at the central interface.

Finally, based on our structural, bioinformatic, and biochemical analyses of the seYdjB-like protein, we constructed a full-length dimer model that includes the TM domain (Fig 5d). Given that this protein appears to contain a TM domain, it is likely a membrane protein. When the final model is positioned within the membrane, it adopts a unique X-shaped dimeric structure (Fig 5e). Based on our findings, we conclude that the seYdjB-like protein might be a membrane protein containing a noncanonical ACT-like domain that functions as a dimer with active sites located at the center of the dimer interface. Further studies will be needed to elucidate its precise biological function.

In this study, we analyzed the YdjB-like protein from *Salmonella enterica*, which appears to be a putative dimeric membrane-associated protein. Our structural analysis, combined with bioinformatic predictions, revealed that it adopts a fold that is completely distinct from the canonical YdjB structure. Despite these intriguing findings, the absence of data on cellular localization, biochemical characterization, or *in vivo* functional analysis highlights the need for future studies to investigate the physiological role of this uncharacterized protein

## Acknowledgments

We thank the staff at the 5C beamline at the Pohang Accelerator Laboratory (Pohang, Korea) for their assistance with data collection.

## Author contributions

**Conceptualization:** Hyun Ho Park.

**Data curation:** Ju Hyeong Kim, Yong Jun Kang, Young Woo Kang, Hyo Been Jin, Eunmi Hong.

**Formal analysis:** Ju Hyeong Kim, Yong Jun Kang, Young Woo Kang, So Eun Park.

**Funding acquisition:** Hyun Ho Park.

**Project administration:** Hyun Ho Park.

**Supervision:** Hyun Ho Park.

**Writing – original draft:** Ju Hyeong Kim, Hyun Ho Park.

**Writing – review & editing:** Hyun Ho Park.

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
