## [Decision Letter · Decision Letter 0]

15 Sep 2025

Dear Dr. Park,

Thank you for submitting your manuscript to PLOS ONE. After careful consideration, we feel that it has merit but does not fully meet PLOS ONE’s publication criteria as it currently stands. Therefore, we invite you to submit a revised version of the manuscript that addresses the points raised during the review process.

We look forward to receiving your revised manuscript.

Kind regards,

Georgios Pantouris, Ph.D.

Academic Editor

PLOS ONE

Journal Requirements:

https://www.sciencedirect.com/science/article/abs/pii/S0006291X25008654?via%3Dihub

In your revision ensure you cite all your sources (including your own works), and quote or rephrase any duplicated text outside the methods section. Further consideration is dependent on these concerns being addressed.

“. This study was supported by the National Research Foundation of Korea (NRF) grant funded by the Korea government (MSIT) (RS-2025-02316334 and RS-2024-00344154).”

“. This study was supported by the National Research Foundation of Korea (NRF) grant funded by the Korea government (MSIT) (RS-2025-02316334 and RS-2024-00344154).”

“We thank the staff at the 5C beamline at the Pohang Accelerator Laboratory (Pohang, Korea) for their assistance with data collection. This study was supported by the National Research Foundation of Korea (NRF) grant funded by the Korea government (MSIT) (RS-2025-02316334 and RS-2024-00344154).”

“. This study was supported by the National Research Foundation of Korea (NRF) grant funded by the Korea government (MSIT) (RS-2025-02316334 and RS-2024-00344154).”

6. Please note that your Data Availability Statement is currently missing the DOI/accession number of each dataset OR a direct link to access each database. If your manuscript is accepted for publication, you will be asked to provide these details on a very short timeline. We therefore suggest that you provide this information now, though we will not hold up the peer review process if you are unable.

7. PLOS ONE now requires that authors provide the original uncropped and unadjusted images underlying all blot or gel results reported in a submission’s figures or Supporting Information files. This policy and the journal’s other requirements for blot/gel reporting and figure preparation are described in detail at https://journals.plos.org/plosone/s/figures#loc-blot-and-gel-reporting-requirements and https://journals.plos.org/plosone/s/figures#loc-preparing-figures-from-image-files. When you submit your revised manuscript, please ensure that your figures adhere fully to these guidelines and provide the original underlying images for all blot or gel data reported in your submission. See the following link for instructions on providing the original image data: https://journals.plos.org/plosone/s/figures#loc-original-images-for-blots-and-gels.

Reviewers' comments:

Reviewer's Responses to Questions

**Comments to the Author**

1. Is the manuscript technically sound, and do the data support the conclusions?

Reviewer #1: Yes

Reviewer #2: Partly

2. Has the statistical analysis been performed appropriately and rigorously?

Reviewer #1: Yes

Reviewer #2: N/A

3. Have the authors made all data underlying the findings in their manuscript fully available?

Reviewer #1: Yes

Reviewer #2: Yes

4. Is the manuscript presented in an intelligible fashion and written in standard English?

Reviewer #1: Yes

Reviewer #2: Yes

Reviewer #1: 1. The functional role of the protein remains confusing. No biochemical or in vivo assays were performed to validate the proposed enzymatic or regulatory function.

2. The annotation as “YdjB-like” is potentially confusing, as the protein is distinct from the known nicotinamidase YdjB (now PncA).

3. The evolutionary and physiological relevance of the protein is not explored beyond structural conservation.

4. The functional implications of the conserved groove or dimer interface are not experimentally explored.

5. The membrane localization model is speculative and not supported by cellular localization data.

6. There are minor grammatical issues and awkward phrasing (e.g., “the structure is somewhat similar to aspartokinase at first glance…”).

7. The title is overly long and could be more concise while retaining key information.

Reviewer #2: This is an interesting article, I have some considerations and questions.

Fist E.coli was used for transformation, however, it's know that coli has its own YdjB-like protein, there's any assay to support no background interference or Protein-Protein interactions?

Please, clarify in the paper why kanamycin was used? There is a resistant sequency in the plasmid?

There is any reason that only the soluble region was used and the N-terminal transmembrane region was excluded?

There is some hypothesis, literature based, for the functional Implications?

There is experimental evidence for membrane association or only bioinformatic predictions?

Should the methodology strengths and weaknesses be cited in the article?

**Do you want your identity to be public for this peer review?** For information about this choice, including consent withdrawal, please see our Privacy Policy

Reviewer #1: **Yes: ** Adithya Polasa

Reviewer #2: No

---

## [Author Response · Author response to Decision Letter 1]

20 Sep 2025

Dear Editor,

We have revised our manuscript based on the reviewers’ suggestions.

Thank you very much for your time and effort in editing our manuscript, and we hope that it is now suitable for publication in “Plos One”.

Sincerely,

Hyun Ho Park, Ph.D.

Professor of Pharmacy

Reviewers' Comments to Author:

Reviewer 1:

We thank the reviewer for the constructive comments on our work.

Comments:

1. The functional role of the protein remains confusing. No biochemical or in vivo assays were performed to validate the proposed enzymatic or regulatory function.

Response: We agree with the reviewer’s comment. Our current manuscript provides new structural insights into a YdjB-like protein. Although the in vivo function and enzymatic properties of this uncharacterized protein remain unknown, we believe these are important topics for future investigation. At present, due to the lack of any existing functional information, further studies must accumulate before reliable functional predictions can be made. At that point, biochemical and in vivo studies will indeed be essential. Nevertheless, we believe that our present work, as the first to report structural features of this functionally uncharacterized protein, offers meaningful and valuable information.

2. The annotation as “YdjB-like” is potentially confusing, as the protein is distinct from the known nicotinamidase YdjB (now PncA).

Response: Yes, that is correct. We also agree with this comment from the reviewer. The name “YdjB-like protein” was not assigned by our research team; rather, it is the name currently listed in major gene annotation databases such as UniProt. Therefore, we specifically emphasized in our manuscript that, despite the name, this YdjB-like protein is structurally distinct from YdjB.

4. The functional implications of the conserved groove or dimer interface are not experimentally explored.

Response: As mentioned in our response to Comment 1, we agree that if functional studies on this protein are to be conducted in the future, this would indeed be an essential and important research direction. We sincerely thank the reviewer for providing this valuable idea.

5. The membrane localization model is speculative and not supported by cellular localization data.

Response: We fully agree that this topic ultimately needs to be validated in a cellular context. However, we would like to reiterate that the focus of our current study is the structural characterization of this protein, highlighting its novel architecture and structural differences from YdjB. In addition, we present preliminary bioinformatics data suggesting that this uncharacterized protein may localize to the membrane, despite the lack of known function. Our laboratory is currently conducting a detailed functional analysis of this protein, and we believe that the suggestions made by the reviewer will be highly valuable for that future work. Once again, we sincerely thank the reviewer for the many insightful research suggestions.

6. There are minor grammatical issues and awkward phrasing (e.g., “the structure is somewhat similar to aspartokinase at first glance…”).

Response: Based on the reviewer’s suggestion, we have modified the sentence to make the meaning more precise and scientifically clear.

7. The title is overly long and could be more concise while retaining key information.

Response: We appreciate the reviewer’s suggestion. We also agreed that the original title was too long. In response to the comment, we have revised the title to make it more concise.

The new title is: A putative membrane-associated YdjB-like protein from Salmonella enterica exhibits a non-canonical ACT-like fold.

Reviewer 2:

We thank the reviewer for the constructive comments on our work.

Comments:

1. Fist E.coli was used for transformation, however, it's know that coli has its own YdjB-like protein, there's any assay to support no background interference or Protein-Protein interactions?

Response: We understand the reviewer’s concern. However, the YdjB-like protein used in our study was overexpressed and purified from E. coli, allowing us to obtain sufficient amounts of the target protein without being limited by the endogenous YdjB-like protein expressed by E. coli.

2. Please, clarify in the paper why kanamycin was used? There is a resistant sequency in the plasmid?

Response: Yes, that is correct. The plasmid vector used in our study carries a kanamycin resistance gene.

3. There is any reason that only the soluble region was used and the N-terminal transmembrane region was excluded?

Response: Thank you for the insightful question. In structural studies, it is often necessary to obtain large amounts of soluble protein, which can be particularly challenging for membrane proteins. To address this, it is a common strategy to remove predicted transmembrane domains and focus on the soluble regions for structural characterization. In our study, we followed this approach by removing the N-terminal region predicted to contain the transmembrane domain. This allowed us to successfully express, purify, and determine the structure of the soluble portion of the protein.

4. There is some hypothesis, literature based, for the functional Implications?

Response: This protein was initially annotated as a “YdjB-like protein” based on predicted sequence and structural similarity to YdjB. However, as demonstrated in our current study, its structure is entirely different from that of YdjB. This structural divergence suggests that the protein may have a distinct function. To our knowledge, there are currently no published studies describing the biological or biochemical function of this protein.

5. There is experimental evidence for membrane association or only bioinformatic predictions?

Response: Our study does not include experimental validation; it only presents bioinformatic predictions. We fully agree that this topic ultimately needs to be validated in a cellular context. However, we would like to reiterate that the focus of our current study is the structural characterization of this protein, highlighting its novel architecture and structural differences from YdjB. In addition, we present preliminary bioinformatics data suggesting that this uncharacterized protein may localize to the membrane, despite the lack of known function. Our laboratory is currently conducting a detailed functional analysis of this protein, and we believe that the suggestions made by the reviewer will be highly valuable for that future work. Once again, we sincerely thank the reviewer for the many insightful research suggestions.

6. Should the methodology strengths and weaknesses be cited in the article?

Response: We believe this is an excellent suggestion. In response to the reviewer’s comment, we have added a new paragraph at the end of the “Results and Discussion” section to address both the methodological strengths and limitations of our study, as outlined below.

“In this study, we analyzed the YdjB-like protein from Salmonella enterica, which appears to be a putative dimeric membrane-associated protein. Our structural analysis, combined with bioinformatic predictions, revealed that it adopts a fold that is completely distinct from the canonical YdjB structure. Despite these intriguing findings, the absence of data on cellular localization, biochemical characterization, or in vivo functional analysis highlights the need for future studies to investigate the physiological role of this uncharacterized protein”

Journal Requirements:

1. Please ensure that your manuscript meets PLOS ONE's style requirements, including those for file naming. The PLOS ONE style templates can be found at https://journals.plos.org/plosone/s/file?id=wjVg/PLOSOne_formatting_sample_main_body.pdf

And https://journals.plos.org/plosone/s/file?id=ba62/PLOSOne_formatting_sample_title_authors_affiliations.pdf

Response: Done

https://www.sciencedirect.com/science/article/abs/pii/S0006291X25008654?via%3Dihub

In your revision ensure you cite all your sources (including your own works), and quote or rephrase any duplicated text outside the methods section. Further consideration is dependent on these concerns being addressed.

Response: We did.

“. This study was supported by the National Research Foundation of Korea (NRF) grant funded by the Korea government (MSIT) (RS-2025-02316334 and RS-2024-00344154).”

Response: Done

“. This study was supported by the National Research Foundation of Korea (NRF) grant funded by the Korea government (MSIT) (RS-2025-02316334 and RS-2024-00344154).”

Response: Done

“We thank the staff at the 5C beamline at the Pohang Accelerator Laboratory (Pohang, Korea) for their assistance with data collection. This study was supported by the National Research Foundation of Korea (NRF) grant funded by the Korea government (MSIT) (RS-2025-02316334 and RS-2024-00344154).”

“. This study was supported by the National Research Foundation of Korea (NRF) grant funded by the Korea government (MSIT) (RS-2025-02316334 and RS-2024-00344154).”

Response: Done

6. Please note that your Data Availability Statement is currently missing the DOI/accession number of each dataset OR a direct link to access each database. If your manuscript is accepted for publication, you will be asked to provide these details on a very short timeline. We therefore suggest that you provide this information now, though we will not hold up the peer review process if you are unable.

Response: Included

7. PLOS ONE now requires that authors provide the original uncropped and unadjusted images underlying all blot or gel results reported in a submission’s figures or Supporting Information files. This policy and the journal’s other requirements for blot/gel reporting and figure preparation are described in detail at https://journals.plos.org/plosone/s/figures#loc-blot-and-gel-reporting-requirements and https://journals.plos.org/plosone/s/figures#loc-preparing-figures-from-image-files. When you submit your revised manuscript, please ensure that your figures adhere fully to these guidelines and provide the original underlying images for all blot or gel data reported in your submission. See the following link for instructions on providing the original image data: https://journals.plos.org/plosone/s/figures#loc-original-images-for-blots-and-gels.

Response: We used uncropped gel in the main figures.

Response: Ok

Response: Confirmed

---

## [Editor Report · Decision Letter 1]

30 Sep 2025

Dear Dr. Park,

Please submit your revised manuscript by Nov 14 2025 11:59PM. If you will need more time than this to complete your revisions, please reply to this message or contact the journal office at plosone@plos.org . A rebuttal letter that responds to each point raised by the academic editor and reviewer(s). You should upload this letter as a separate file labeled 'Response to Reviewers'.A marked-up copy of your manuscript that highlights changes made to the original version. You should upload this as a separate file labeled 'Revised Manuscript with Track Changes'.An unmarked version of your revised paper without tracked changes. You should upload this as a separate file labeled 'Manuscript'.

We look forward to receiving your revised manuscript.

Kind regards,

Georgios Pantouris, Ph.D.

Academic Editor

PLOS ONE
---

## [Author Response · Author response to Decision Letter 2]

14 Oct 2025

Comment: For transparency, please include the wwPDB validation report generated for your structure. 

Response: We included the PDB validation report for our structure as requested.

---

## [Editor Report · Decision Letter 2]

26 Oct 2025

A putative membrane-associated YdjB-like protein from Salmonella enterica exhibits a non-canonical ACT-like fold

PONE-D-25-45368R2

Dear Dr. Park,

We’re pleased to inform you that your manuscript has been judged scientifically suitable for publication and will be formally accepted for publication once it meets all outstanding technical requirements.

Kind regards,

Georgios Pantouris, Ph.D.

Academic Editor

PLOS ONE

---

## [Editor Report · Acceptance letter]

PONE-D-25-45368R2

PLOS ONE

Dear Dr. Park,

I'm pleased to inform you that your manuscript has been deemed suitable for publication in PLOS ONE. Congratulations! Your manuscript is now being handed over to our production team.

Kind regards,

on behalf of

Professor Georgios Pantouris

Academic Editor

PLOS ONE